# Protocol of a Single-Blind Two-Arm (Waitlist Control) Parallel-Group Randomised Controlled Pilot Feasibility Study for mHealth App among Incontinent Pregnant Women

**DOI:** 10.3390/ijerph18094792

**Published:** 2021-04-30

**Authors:** Aida Jaffar, Sherina Mohd Sidik, Chai Nien Foo, Noor Azimah Muhammad, Rosliza Abdul Manaf, Siti Irma Fadhilah Ismail, Nazhatussima Suhaili

**Affiliations:** 1Department of Psychiatry, Faculty of Medicine and Health Sciences, Universiti Putra Malaysia, Seri Kembangan 43400, Malaysia; aida@upnm.edu.my (A.J.); irma@upm.edu.my (S.I.F.I.); 2Primary Care Unit, Faculty of Medicine and Defence Health, Universiti Pertahanan Nasional Malaysia, Wilayah Persekutuan Kuala Lumpur 57000, Malaysia; 3Department of Population Medicine, Universiti Tunku Abdul Rahman, Kajang 43000, Malaysia; 4Department of Family Medicine, Faculty of Medicine, Universiti Kebangsaan Malaysia, Wilayah Persekutuan Kuala Lumpur 56000, Malaysia; drazimah@gmail.com; 5Department of Community Health, Faculty of Medicine and Health Sciences, Universiti Putra Malaysia, Seri Kembangan 43400, Malaysia; rosliza_abmanaf@upm.edu.my; 6Klinik Kesihatan Ampang, Ampang 68000, Malaysia; drshima.suhaili@gmail.com

**Keywords:** mHealth app, mobile application, pelvic floor muscle training, urinary incontinence, pregnancy, behavioural change theory, randomized control trial, pilot feasibility study

## Abstract

Background: The delivery of pelvic floor muscle training (PFMT) through mHealth apps has been shown to produce promising results in improving pelvic floor muscle strength and urinary incontinence (UI). However, there is limited evidence on mHealth apps designed for pregnant women who are at high risk of developing UI. This pilot study aims to evaluate the feasibility of conducting an effectiveness trial for a newly developed PFMT app among pregnant women in Malaysia. Methods: This is a prospective, single-centre, single-blind, randomised controlled pilot feasibility study: The Kegel Exercise Pregnancy Training app (KEPT-app) Trial. Sixty-four incontinent pregnant women who attended one primary care clinic for the antenatal follow-up will be recruited and randomly assigned to either intervention or waitlist control group. The intervention group will receive the intervention, the KEPT-app developed from the Capability, Opportunity, Motivation-Behaviour (COM-B) theory with Persuasive Technology and Technology Acceptance Model. Discussion: This study will provide a fine-tuning for our future randomised control study on the recruitment feasibility methods, acceptability, feasibility, and usability of the KEPT-app, and the methods to reduce the retention rates among pregnant women with UI. Trial registration: This study was registered on ClinicalTrials.gov on 19 February 2021 (NCT04762433) and is not yet recruiting.

## 1. Introduction

Urinary incontinence (UI) is defined as involuntary urinary leakage [1]. A recent cross-sectional study in a Malaysian primary care centre reported that 40.9% of pregnant women had UI [2], which is similar to a meta-analysis that reported 41.0% (range of 9–75%) of 88,305 pregnant women had UI based on 44 included studies [3].

UI during pregnancy has resulted in slight to moderate impairment of quality of life (QOL) [4]. UI poses a risk of a person’s physical, functional, and psychological morbidities [5]. Pregnant women with UI may experience negatively impacted QOL in terms of their social-emotional relationships, difficulty in doing exercise and sport, employment issues, restrictions when travelling, sleeping disturbances, and difficulty during prayers [6]. This will be expected especially in Malaysia, where it is a Muslim country and to be dry before ablution and during the prayers is mandatory for the five times daily prayers.

Pelvic floor muscle training (PFMT) or Kegel exercise is an essential exercise to strengthen pelvic floor muscles among pregnant women [7]. PFMT during the first pregnancy may help shorten the first and second stage of labour [8]. PFMT during pregnancy can prevent pelvic floor dysfunction, for example, urinary incontinence in late pregnancy and the early postpartum period [9]. Unfortunately, despite affecting their daily activities, only one-tenth (13.1% of 407 pregnant women) seek help due to the misperception that UI would resolve itself [4].

The barriers in performing PFMT among pregnant women can be divided into several factors, for example, normalisation of having UI [10], limited resources [10] that lead to limited knowledge of PFMT [2], and misconception that PFMT will lead to miscarriage [10]. Correct and reliable information on PFMT must be easily accessible via the internet or the mHealth app for the target population.

Acquiring correct PFMT skills may be challenging for pregnant women, resulting in poor adherence [11]. First, they must understand the basic facts about UI, adopt the correct attitude towards PFMT, and learn and practice PFMT from a skilful instructor. Following this, their confidence will be built up to obtain good self-efficacy of PFMT. They will then be able to implement and incorporate PFMT into her daily activities to ensure its retention and sustainability. Therefore, a suitable and well-designed intervention would help motivate pregnant women in adopting PFMT into their daily activities.

There are barriers from health personnel when discussing and providing instructions of PFMT as this is not routinely practiced [11]. These factors will reduce the availability of the services and affect the accessibility of PFMT to pregnant women [12]. However, despite being accessible, the acceptability of PFMT services is crucial to ensure that the pregnant women are willing to learn and adhere with PFMT. Therefore, an innovative way that involves easy, low-risk, and proven strategies may improve the availability, accessibilities, and acceptability of PFMT among pregnant women.

Mobile health (mHealth) is defined as “a medical and public health practice supported by mobile devices, such as mobile phones, patient monitoring devices, personal digital assistants, and other wireless devices” [13]. mHealth has many potential benefits in health behaviour interventions. Its strengths lie in its ease of access and user-friendliness, resulting in its widespread adoption worldwide [14]. Furthermore, pregnant women using the apps will have timing flexibility, save money, and feel less embarrassment [15,16,17].

mHealth apps have shown their effectiveness in self-management of pregnancy, improving healthcare delivery, improving cholesterol levels, improving mood and energy levels of pregnant women, and less hospital visits among the high-risk pregnancy groups [18]. mHealth apps may include some features such as audio guidance for PFMT, which contributes to its effectiveness in improving compliance among pregnant women [16]. Tät app has proven its pragmatic effectiveness for self-management of incontinence among non-pregnant women, which has been downloaded up to 65,000 times during its study period [19]. DiárioSaúde app was designed by utilising operant conditioning among 17 non-pregnant women with an improvement in incontinence reported with PFMT adherence [20]. Hence, PFMT intervention may be delivered via mHealth apps to improve the self-efficacy and adherence towards PFMT among pregnant women with an appropriate implementation strategy [21].

Limitations and concerns of mHealth apps need to be addressed when pregnant women download the apps, such as its content accuracy, risk of confidentiality breaches, privacy and security, and the lack of central regulation and professional involvement in their development [22]. Even though 32 out of 120 apps related to PFMT, which included 15 paid apps and 17 free apps, are available in official stores, only a limited number of PFMT apps related to pregnant women have undergone clinical trials [23]. Professionals and researchers need to develop PFMT apps to ensure pregnant women’s safety and to improve the quality of maternity care.

Successful implementation of PFMT in maternity care is influenced by three crucial factors: (1) personal factors, (2) organisational factors, and (3) contextual factors [24]. Personal factors include the motivation of the pregnant woman, her perception towards the value of the new behaviour, and PFMT skill development [24]. Organisational factors involve stakeholder engagement in adopting the intervention and service provider involvement in the design and development of the intervention [24]. Meanwhile, the contextual factors consist of adapting the intervention to fit into the current maternity service provision and the capacity to accommodate for changes [24]. Subsequently, a pilot study should be able to investigate any barriers or issues from these three factors. 

The pilot study is an essential step before implementing a full-scale phase III trial [25]. Before a full-scale study is conducted, a pilot study can help to assess the feasibility of the key processes of the main study, such as evaluating the recruitment rates. Pilot studies also help in identifying possible problems that may crop up during the main study. These may include time and resource issues or human and data management issues related to implementing the intervention in a busy primary care clinic. Finally, a pilot study can provide information on the preliminary effectiveness and the related processes of the intervention, such as the appropriate duration of daily exercise to have a significant PFMT adherence among pregnant women [25]. 

We will conduct a pilot and feasibility study to assess the feasibility, acceptability, and usability of a PFMT mHealth app. The app is a Kegel Exercise Pregnancy Training app (KEPT-app), which is compatible with Android version, developed by Pheon Tech, Selangor, Malaysia. At the time of writing, KEPT-app is undergoing usability evaluation via heuristic assessment to assess the user interface, cognitive walkthrough to assess the learnability of the app, and think-aloud to assess the user’s experience in using the app. The usability of this app in the clinical setting will be evaluated during this pilot feasibility study.

Initially, we published the study protocol for the KEPT-app trial [26]. The trial would have utilised a cluster RCT design, which would have involved ten primary care clinics. However, due to unforeseen circumstances related to the COVID-19 pandemic, including social restrictions to conduct research at public healthcare facilities, changes were required to conduct the research. Therefore, a smaller scale feasibility trial was planned, focusing on one health clinic instead of the original multicentre trial. This trial was needed to identify possible barriers and modifications required to enable the full study to be conducted as a pragmatic RCT.

This pilot study aims: (1) to examine the acceptability and feasibility of the KEPT-app, to determine the appropriate training time in achieving the advanced skills level; (2) to determine the dropouts from the control group and to assess any dissatisfaction among the control group; (3) to assess recruitment rate, to determine the reasons from dropout participants (if any); and (4) to assess the preliminary effectiveness in knowledge, attitude, practice, and adherence to PFMT, PFMT self-efficacy, PFMT adherence, UI, and quality of life. The cost-effectiveness analysis will be performed to assess the health economy of the KEPT-app from the societal perspectives for treating UI among pregnant women. To our best knowledge, KEPT-app is the first designed app for all types of parity among incontinent pregnant women. Therefore, a pilot feasibility study is crucial to assist in the feasibility of the future KEPT-app trial study

## 2. Materials and Methods

Design overview: This pilot study will use a single-blind, single-centre, parallel, randomised controlled trial (RCT) design that will incorporate individualized treatment flexibility, as in a real-world setting, and provide the future study’s feasibility process [26]. The study protocol was designed and reported according to CONSORT (Consolidated Standards of Reporting Trials) extension for randomised pilot and feasibility trials [27,28].

Participants (eligibility criteria): The eligible participants are: (1) pregnant women aged over 18 years old, (2) any parity at 26–27 weeks’ gestation, (3) with either stress UI or mixed UI according to the International Consultation on Incontinence Questionnaire—UI Short Form [29,30], and (4) Malaysian citizens. Other citizens will be excluded as the primary language used in this study will be Malay, the national language of Malaysia.

Additionally, pregnant women with chronic medical problem(s) before pregnancy, complicated pregnancies, or conditions with which it is not advisable to practise PFMT will be excluded.

Recruitment and withdrawal: This study will be conducted at one primary care clinic in the Hulu Langat district under the same management as the future study [26]. The recruitment process will follow the future study, whereby pregnant women will be invited from a poster announcement [26]. They will then contact the research assistant (RA) via WhatsApp or Short Message System (SMS) for the eligibility screening process. Once a pregnant woman is eligible to join the study, the RA will provide her with a link to download the app for the consent forms. (Figure 1)

All study participants can withdraw at any time during the trial with or without any reasons. The RA will contact those who have withdrawn to obtain reasons and their experiences joining this pilot study. This finding will be analysed and given as input into the barriers that will add to future study modifications.

For the sample size, an analysis suggested that 12 participants per group is an appropriate sample in a feasibility study [31] and another review stated that 15 participants per group could be adequate for moderate effect sizes [32]. A pilot randomised controlled trial for an expected small effect size should have 25 participants per arm [33]. However, considering a possible 20% dropout rate, the final sample size will be 64 participants with 32 participants per arm.

Regarding the randomisation and blinding, a random allocation will be applied to ensure the balance between the intervention and control groups. The RA will perform the stratified randomisation with two lists of identification numbers with primigravida or multigravida, because gravida is a significant prognostic risk factor in UI [34]. After the two lists have been made, the RA will use permuted blocks of two or four via RRApp, a randomisation app [35] for random allocation into the intervention and control group. The assigned group allocations will be concealed in an opaque envelope assigned by the nonresearchers in this study. Although study participants will not be blinded, both RA and the researchers who are performing statistical analysis will be blinded to the study participants’ allocation to fulfil the single-blind study criteria [36]. 

Study participants will receive the intervention (KEPT-app) to improve their pelvic floor muscle training in addition to their usual antenatal care. KEPT-app is designed to educate pregnant women on PFMT via step-by-step training with the capability element to improve their confidence and skills (beginner, intermediate, and advance skills). Pregnant women are encouraged to continue to adopt PFMT throughout pregnancy until the postpartum period. The details of the mHealth app content were developed based on the need assessment gathered from the cross-sectional study [2]. It was conducted to identify the need for PFMT and understand the knowledge, attitude, and practices towards PFMT.

KEPT-app is derived from a comprehensive theoretical framework that is Capability, Opportunity, and Motivation-Behaviour (COM-B) model and delivered using the persuasive technology (PT) approach [37,38]. The COM-B model is an extension and assimilation of 14 domains from the Theoretical Domains Framework (TDF) [39]. An additional theory for the KEPT-app will be the Technology Acceptance Model, whereby the usefulness and the ease of use will increase the respondents’ intention to continue using the app [40].

Perceived usefulness refers to “the degree to which a person believes that using a particular system would enhance his or her job performance” and perceived ease of use indicates “the degree to which a person believes that using a particularly stem would be free of effort” [40]. Meta-analysis highlighted that perceived usefulness is the most common predictor of mHealth use and showed a medium-strength relationship with perceived ease of use [41]. Therefore, the mHealth app with a simple design will encourage the users to use it, and their perception of effortless use the app will motivate them to continue using the app. The relationship between Technology Acceptance Model (TAM), Persuasive Technology, and COM-B model is illustrated in Figure 2.

The KEPT-app, developed for the Android version 9.0 operating system, consists of the following main user interfaces: (1) the main login page, (2) the PFMT training video, (3) the exercise module with a timer, (4) a calendar for users to chart their wet days and completed exercise, and (5) a score board. At the time of writing, the KEPT-app prototype is undergoing usability evaluation by both the experts in informative technology and health informatics and from the end-users (pregnant women with UI). Feedback from the usability evaluation will be used to revise the app. The final KEPT-app will then be tested during this feasibility study. The app will be delivered only to the study participants and not through the official stores.

The control group will receive antenatal care as usual during the intervention period and access to install the app after 36 weeks of pregnancy. Routine antenatal care follow-up will be based on the Ministry of Health guidelines [42]. 

Primary outcomes are the outcomes relevant to the acceptability of the KEPT-app. Acceptability and feasibility of the KEPT-app is the satisfaction and agreeableness of the app’s pregnant women [43]. Acceptability is crucial as it affects the success of an implementation phase [44]. The acceptability will be assessed for its content, complexity, and comfort [43]. 

The KEPT-app feasibility will be defined as KEPT-app’s extent of successfully being used or carried out in a primary care clinic setting [45]. The app feasibility will be assessed along with the usability assessment, which can assess the ease of use, interface, satisfaction, and mHealth app usefulness [46]. Additionally, the PFMT education video embedded in the app should assess its understandability and actionability [47]. 

Secondary outcomes are as follows; (1) feasibility of recruiting the study participants, which include numbers and proportions of participants recruited, screened, consented, and numbers with their reasons for withdrawals; (2) face validation on the educational video embedded in the app; (3) appropriate training time in achieving the advanced skill level; (4) control group’s barriers continuing the study; (5) barriers and suggestions for using the app; (6) the knowledge, attitude, and practice of PFMT; and (7) quality of life.

Training time will be recorded automatically via the app and a reminder will be delivered (during the evening) if no PFMT is recorded on that day. There will be three training skills: beginner (2 s each contraction), intermediate (6 s each contraction), and (10 s each contraction). The pregnant women are scheduled for ten repetitions each set and must complete three sets every day. The time recorded will be analysed using the algorithm to measure which training skills will change their self-efficacy and adherence.

It is essential to determine the KEPT-app’s preliminary effectiveness to improve the PFMT skills of pregnant women. The time recorded by the pregnant women will be analysed with the self-efficacy improvements and adherence towards PFMT. Additionally, the recorded time will be analysed with the UI changes within the group.

Additional qualitative data collection will be used to further understand any barriers or suggestions from the study participants. The expenditures during the study, for example, the cost of absorbent pads, participant’s salary, and cost of laundry where applicable, will be compared between the groups to assess the cost-effectiveness.

Trial feasibility (treatment fidelity) is crucial to ensure that PFMT is performed by the participants according to the system log recorded. The monitoring must be continuously performed to ensure the treatment fidelity. Treatment fidelity describes “the methodological strategies used to monitor and enhance the reliability and validity of behavioural interventions” [48]. Therefore, the RA must actively monitor the system log activities (training time) and immediately inform the researcher if any system failure or no activities are recorded by the patients.

The adherence to the training time is when the pregnant women complete 80% of the expected training time, which will be continuously recorded after they watch the video [49]. The calendar apps method will record their PFMT, and if they have not logged the data, a reminder will be delivered automatically. The adherence rate data will inform future studies when the best time is to send the booster or reminders. Any errors or issues will be recorded, and the system update or debugging will be recorded for further analysis.

To assess the trial feasibility of the recruitment of study participants and retention rates, the total number of study participants who can be recruited, screened, and interested in participating within one month will be examined. The researchers will assess the study feasibility by reviewing the timing log data recorded during the pilot study. This recruitment process is critical as there will be no researcher on-site to promote this study due to the uncertainty of the restricted movement due to the COVID-19 pandemic. 

Any dropout or defaulter will be traced and contacted. The number of withdrawals will be recorded and descriptively analysed. There will be no further analysis for the dropouts with per-protocol (PP) analysis and intention-to-treat (ITT) analysis to evaluate the influence of missing data as this is not an effectiveness study.

We will perform primary and secondary outcome measures at three time points during the trial to assess the trial feasibility in collection of outcome data—at enrolment into the study, a baseline assessment will be recorded, and one-month and two-months post-intervention, an assessment will be recorded accordingly for both groups.

The participant feedback towards this app will be assessed via qualitative data collection after the study ends. This feedback will provide the input regarding the trial feasibility based on the barriers to the implementation of the intervention. Pregnant women will add their opinion on the barriers and difficulties and their suggestions in improving this app. 

### 2.1. Data Collection

Study participant demographic data (age, nationality, body mass index (BMI), parity, weeks of gestation, type of UI, and severity) will be recorded on the first day of enrolment into the study. The UI will be assessed via the International Consultation on Incontinence Questionnaire—Urinary Incontinence Short Form (ICIQ-UI SF) [29,30] (Table 1). 

The Technology Acceptance Model (TAM) framework will be used to assess the acceptability of the KEPT-app. According to TAM, the questions assess the complexity, content, and comfort [43], for example, (1) How many stars would you use to recommend this app to your friends? (2) How many stars would you rate the content of this app? (3) How many stars would you rate the comfort using this app? (4) Do you find it complex using this app?

The Malay-mHealth App Usability Questionnaire (M-MAUQ) will be used [46,50] to evaluate the feasibility and usability of the app. This questionnaire will assess the ease of use, interface, satisfaction, and mHealth app usefulness to the study participants. 

The study participants will assess their knowledge, attitude and practices toward PFMT to evaluate any improvement after the intervention [53]. For assessing the self-efficacy and adherence towards PFMT, the Exercise Adherence Rating Scale (EARS) [51] and Self-Efficacy Scale For Practicing Pelvic Floor Exercise Questionnaire (SESPPFE) [52] will be used. Additionally, cost efficiency will be assessed as explained in the study protocol published elsewhere using the quality of life of the study participants [26]. The Patient Educational Material Assessment Tool (PEMAT) will be used to validate the educational video embedded in the app [47].

Responses from the open-ended questions will be analysed further to understand the study feasibility and app acceptability. The participants’ information regarding the recruitment and study process will be assessed, and study process improvement will be taken for the future definitive KEPT-app trial. The consensus of the participants’ technical findings will inform the developer in improving the technical part of the app.

In evaluating the cost analysis study, the data from the two-month follow-up will be extrapolated to one year for the calculations. We will collect each participant’s training time; incontinence aids, for example absorbent pads; and laundry (if applicable) at baseline and at a two-month follow-up among all the participants [54]. We will utilise the International Consultation on Incontinence Modular Questionnaire on Lower Urinary Tract Symptoms and Quality of Life to measure their quality of life to calculate the quality-adjusted life years (QALYs) gained [54].

### 2.2. Statistical Analysis

The primary analysis of this pilot study will be by descriptive analysis to concentrate on the feasibility outcomes. The recruitment number and the proportion of participants screened, consented, and randomised will be presented. Additionally, the information of the dropouts will be presented with their reasons in the results.

The results of study outcomes and validation of educational video will be analysed. Exploratory analysis of the effect sizes for PFMT adherence will be conducted by mixed-effects regression analyses, controlling for significant baseline characteristics, such as age, parity, educational status, and income status. For the cost-effectiveness analysis, expenditures during the study will be compared between the groups using the multiway sensitivity analysis.

As the data will be sent via the app, the system will collect the data and store it in a secured and encrypted cloud storage. The researcher (blinded) who has been trained with a statistical analysis program will analyse the data using the Statistical Package for the Social Sciences version 27.0, IBM, New York, United States of America [55,56].

## 3. Discussion

Living and experiencing UI affects the quality of life among nonpregnant and pregnant women [3,57]. Even though no studies have explored the economic burden of UI during pregnancy [58], UI among women has been stated as a high economic burden as it affects a person psychologically, socially, and financially [59]. Pregnant women face barriers seeking help by assuming that UI “normally” occurs during pregnancy and UI will resolve after giving birth. Hence, an effective intervention is essential to deliver the correct conservative management for UI during pregnancy, either pelvic floor muscle training (PFMT) or Kegel exercise.

KEPT-app trial is a single-blind, pragmatic, cluster RCT that will be implemented with this pilot test’s findings. Being a pragmatic study, which is recommended as it is near to the real current situation, challenges of uncertainty must be well informed. This pilot trial aims to identify the potential challenges of recruitment from one single centre before involving multiple centres for the actual trial. There are various restrictions in collecting data from multiple centres due to the COVID-19 pandemic. Information from a single centre feasibility trial will provide valuable information on whether it is suitable to proceed with data collection from multiple study sites. Therefore, alternatives need to be planned and prepared. Any amendments to the full study protocol will be informed via an erratum to the previously published phase III study protocol [26].

This pilot study will evaluate innovative features of the app such as an embedded electronic consent form. Obtaining informed consent from participants is an essential research ethics process in conducting studies. This consent form will be different from the usual online surveys where it requires the users to sign on the app. The act of physically signing the form is a form of persuasive technique [60], which may contribute towards their commitment towards the study. This electronic consent form may be an alternative process for obtaining participants’ informed consent in future studies where face-to-face methods are not possible.

Additional strengths of this study include evaluating the feasibility and challenges of implementing mobile apps in the maternal and child health setting in a primary care clinic. These questions are vital to be answered with evidence from the results of this pilot study.

The limitation of this study will be that it only evaluates the feasibility of the study from a single centre. There may be other additional implementation findings when the full study is conducted over multiple centres. However, this issue cannot be avoided due to the various restrictions related to conducting research in health facilities during the COVID-19 pandemic. 

## 4. Conclusions

As a conclusion, this study will hopefully understand the feasibility and acceptability of this mHealth app and the intervention study design for future full-scale clinical trials.

## Figures and Tables

**Figure 1 ijerph-18-04792-f001:**
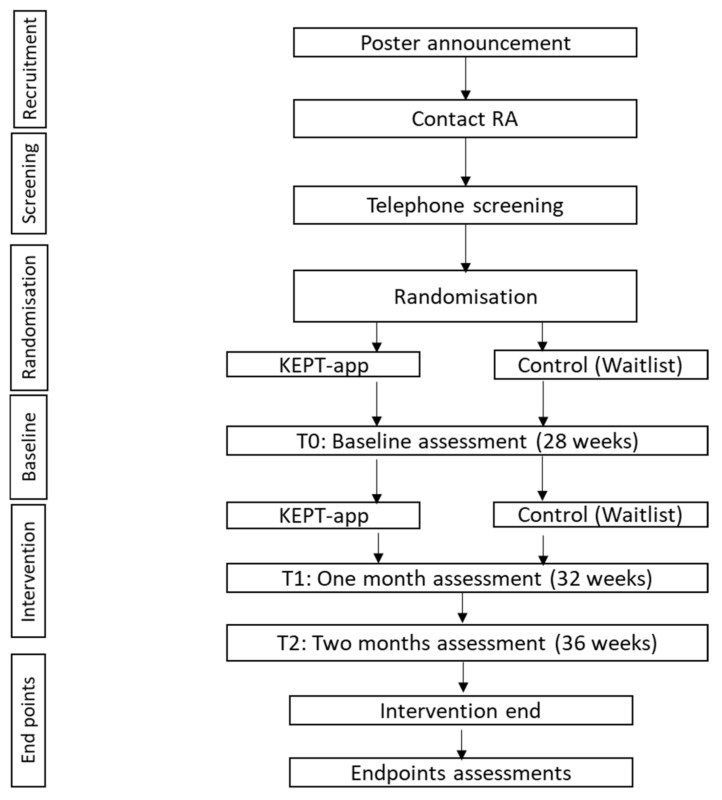
Pilot study flowchart.

**Figure 2 ijerph-18-04792-f002:**
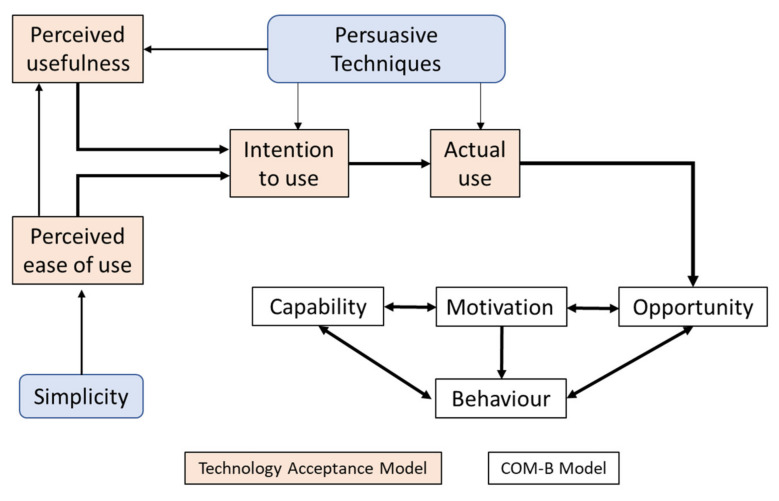
Conceptual framework of KEPT-app.

**Table 1 ijerph-18-04792-t001:** Study outcomes of the KEPT-app trial.

Primary Outcome	Information
Acceptability and feasibility (usability) of the app.	The acceptability will be assessed for its content, comfort, complexity, and other factors. (1) How many stars would you use to recommend this app to your friends? (2) How many stars would you rate the content of this app? (3) How many stars would you rate the comfort using this app? (4) How many stars would you rate simplicity using this app? (5) How many stars would you rate the recruitment process using this app? (6) How many stars would you rate the quality of the app? (7) Factors influenced/hindered using this app? (8) What motivates you to continue the PFMT? (Further information will be collected via qualitative data feedback) Malay-mHealth App Usability Questionnaire (M-MAUQ) will be used [46,50] to assess the ease of use, interface, satisfaction, and the usefulness of mHealth apps to the end user.
**Secondary Outcomes**	**Information**
The log system records.	The recruitment rates, dropout rates from the registry, and the log system records.
International Consultation on Incontinence Questionnaire—Urinary Incontinence Short Form (ICIQ-UI SF) [29,30].	To measure the urinary incontinence score at baseline, one-month, and two-months post-interventions.
Exercise Adherence Rating Scale (EARS) [51]	Increasing PFMT adherence from lowest score (0) to maximum score (24) of Exercise Adherence Rating Scale (EARS). To measure the adherence score at baseline, one-month, and two-months post-interventions.
Self-Efficacy Scale For Practicing Pelvic Floor Exercise Questionnaire (SESPPFE) [52]	Improving self-efficacy by scores of higher than 70% of the self-efficacy scale for practicing pelvic floor exercise questionnaire (SESPPFE). To measure the self-efficacy score at baseline, one-month, and two-months post-interventions.
Patient Education Materials Assessment Tool (PEMAT) [47].	To assess the understandability and actionability of the video. To validate the video at one-month post intervention.
Knowledge, Attitude, and Practice towards Pelvic Floor Muscle Training [53]	To assess the knowledge, attitude, and practices towards PFMT at baseline, one-month, and two-months post-interventions.
International Consultation on Incontinence Questionnaire Urinary Incontinence—Lower Urinary Tract Symptom quality of life (ICIQ-LUTSqol) [29,30].	To assess the quality of life among pregnant women with UI at baseline, one-month, and two-months post-interventions.

## Data Availability

Not applicable.

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
