# Peer review of "Protocol of a Single-Blind Two-Arm (Waitlist Control) Parallel-Group Randomised Controlled Pilot Feasibility Study for mHealth App among Incontinent Pregnant Women"

_ijerph, 2021, doi:10.3390/ijerph18094792_

Round 1

Reviewer 1 Report

General comments:

  • The manuscript would benefit from a thorough English check, including grammar and phrasing. This would improve readability and clarity of meaning.
    • A few of examples of where this can be improved:
      • 45: “Urinary incontinence (UI) is a condition where a person is unable to control urine, leading to its involuntary leakage.”
      • 50: “During pregnancy, UI significantly affects the quality of life, with a review reported that 90% of them have a slight to moderate impact on the quality of life.”
      • 65: “A pregnant woman faces challenges in adhering to PFMT due to its complexity in acquiring PFMT skills.”
      • 78: “The pilot feasibility study is an important step to ensure its feasibility and acceptability before implementing a full-powered randomised control trial.”
  • How is this pilot protocol related to the cluster trial protocol recently published in BMJ Open? (http://dx.doi.org/10. 1136/bmjopen-2020-039076). If this pilot protocol was intended to inform the design of the fully powered study (particularly sample size), why is the full study protocol already published?
    • This manuscript states that the pilot is not yet recruiting, supported by the ANZCTR registration. However, the NCT trial registration that was last updated earlier in March 2021 states that the full trial is intended to commence in April 2021. Are the authors intending to run the pilot and the cluster RCT concurrently? Or is this pilot study completed? Please clarify.
    • Pilot study was registered on ANZCTR in Feb 2021 and full study protocol was accepted for publication by BMJ Open in December 2020?

Abstract:

21/22: “Pelvic floor muscle training (PFMT) or Kegel exercise delivered via the mHealth app has shown promising results in improving pelvic floor muscle strength and urinary incontinence (UI).” – This is a confusing opening statement; it sounds more like a concluding statement. Are the authors referring to existing trials in the area? If so, this needs to be rephrased, something like: “The literature indicates that delivery of PFMT (or Kegel exercise) through mHealth apps produces promising results…”

30: Missing “of” after “objective”

30: Suggest moving the statement of objective into the “background” section just before the methods. Additionally, it would be good to also mention the aim to determine feasibility of the future trial. This will keep things orderly and easy to follow.

Introduction:

47: remove the word “almost”, it isn’t necessary as the two percentages are similar.

In the abstract the authors mention that delivery of PMFT through apps has shown promising results, but there us little evidence for mHealth apps in pregnant women. This information does not appear to be referenced or discussed anywhere in the introduction and would provide sound rationale for the author’s study.

Are there any references to existing literature to support the statements made in paragraph commencing line 65?

72: Perhaps mention that implementation relates to maternity care here. At present the sentence is unspecific.

Paragraph commencing line 72 outlines three crucial factors for successful implementation. There is no reference for the sentence starting line 76; and the second listed factor “organisational factors” is not expanded upon.

84: Has the app already been launched? Is it in use in a clinical environment/available in app stores? Has the heuristic testing been completed, or it will be launched after the testing? This is another instance where an English check would be helpful, I’m not sure of the current status based on this description.

The last paragraph of the introduction appears to be slightly repetitive. Lines 83-87 are fragmented and do not link together very well, however, lines 88-97 are clearer and easier to follow the logic of what is to come with the pilot study.

Overall, the introduction for this protocol is very short and appears spliced together. The introduction for the authors’ other recently published protocol in BMJ Open is far more developed.

Methods:

101: “estimation” of what? Does this mean the pilot study will enable authors to assess if the future RCT is feasible?

102: “The study protocol is designed and reported in the CONSORT…” do the authors mean “…according to the CONSORT…”?

COM-B and Tam are appropriate frameworks for this work.

170: This sentence starts to answer my earlier question about how far along the app development is at this point. However, I think it would be very useful to clearly state early on in the methods exactly what point the app development is at, then what level of development it will be at following the completion of the pilot study going into the full RCT.

174: “For the control group, it is similar with the future RCT study, whereby all study…” should the future trial not be similar to this pilot trial, if in fact they are conducted sequentially?

Discussion:

302: This is the first-time cluster is mentioned as part of the study design. Throughout it has been described as a single-centre study. If the main study is to be a cluster, why not trial clustering on a smaller scale in the pilot?

304-307: Do the authors mean that they decided to undertake this pilot after publishing their full trial protocol in order to determine work arounds given COVID restrictions? Is there a plan to update the protocol in BMJ Open with the findings from the pilot study?

308: This information would have been interesting in the introduction.

Conclusion:

Once clarifications are made, it may be prudent to revise the conclusion to be in line.

Reviewer 2 Report

Summary:
The authors of the paper "Parallel-Group, Randomised, Single-centred, Single-blind, Pilot 
and Feasibility Study for pelvic floor muscle training mHealth app: Study protocol" describe the protocol 
of a study which aims to utilize mHealth technology in the context of urinary incontinence during pregnancy.
The authors describe relevant background information, related works and their goals. Based on this, the planned
study is described. This especially includes the framework and setting used to measure the usefulness of the
planned mHealth app. The entire procedure of the study is also explained in great detail. The authors conclude
by mentioning the planned study based on the presented study protocol.

Points in favor:
- The paper is generally written well
- The paper deals with a topical subject
- The paper draws a clear contribution
- The paper shows content that is of broader interest
- The paper describes the study setting in a comprehensive and sound manner

Points against the paper:
- The title is too broad for the specific topic addressed by pelvic floor muscle training
- The aspects of the study are prominently in the title, but not repeated that way in the summary
- Potential and already discovered limitations should be better discussed, including findings from
  the literature when using mHealth technology
- The paragraph in the discussion starting with "A review reported .." contains information that is
  essential for readers, therefore it must be explained already within the Introduction
- Abstract should be improved in general
  E.g.
   "The objective 30 this pilot study is to evaluate the acceptability and feasibility of the KEPT-app."
   Not a correct sentence; in addition, the app is not ready, but is seems that it is already implemented
  -> Other parts of the abstract should also be improved, for example, the first sentence is misleading
- In addition to the latter point, some other sentences pose also minor language issues, please revise the
  entire text again
  E.g.
  Table 1: Study outcomes KEPT-app trial -> Study outcomes of the KEPT-app trial.
  Or,
  Line 250: "According to TAM, the questions which assess the complexity, content and comfort [30] for example;"
  -> "According to TAM, the questions, which assess the complexity, content and comfort are [30], for example;"
- More information about the app should be provided. Planned versions: iOS, Android, web-based
  -> is there any prior work on the app since it will be the central instrument of the study
  -> will the apps be delivered through the official stores or not
  -> will questionnaires be provided through the app for the app arm?
- Is the planned or to be used content already prepared?

Round 2

Reviewer 2 Report

Dear Authors,

two issues remain in my eyes.

(1) The differences to your work [26] must be discussed earlier in the Introduction. In addition, this should be listed in detail and discussed.

(2) Quality of Figures 1 and 2 can be improved, especially how they are scaled.

Author Response

Dear reviewer,

We appreciate your concerns.

(1) The differences to your work [26] must be discussed earlier in the Introduction. In addition, this should be listed in detail and discussed.

Authors:

We have added as below.

L 130

Initially, we have published the study protocol for the KEPT-app trial [26]. The trial was a cluster RCT design which will be involving ten primary care clinics. However, due to unforeseen circumstances related to the COVID-19 pandemic, including social restrictions to conduct research at public healthcare facilities, changes were required to the conduct of the research. Therefore, a smaller scale feasibility trial was planned, focusing on one health clinic instead of the original multi-centre trial. This was needed to identify possible barriers and modifications required to enable the full study to be conducted as a pragmatic RCT.

(2) Quality of Figures 1 and 2 can be improved, especially how they are scaled

Auhtors: We have improved Figure 1 and Figure 2